

# Emerging roles of APLN and APELA in the physiology and pathology of the female reproductive system

Xueying Wang[1,2,*], Xiaofei Liu[1,2,*], Zifan Song[1], Xin Shen[1], Siying Lu[1], Yan Ling[3] and Haibin Kuang[1,4]

[1] Department of Physiology, Basic Medical College, Nanchang University, Nanchang, China
[2] Department of Clinical Medicine, School of Queen Mary, Nanchang University, Nanchang, China
[3] Department of Obstetrics and Gynecology, Jiangxi provincial People's Hospital affiliated Nanchang University, Nanchang, China
[4] Jiangxi Provincial Key Laboratory of Reproductive Physiology and Pathology, Medical Experimental Teaching Center of Nanchang University, Nanchang, China
[*] These authors contributed equally to this work.

## ABSTRACT

APLN, APELA and their common receptor APLNR (composing the apelinergic axis) have been described in various species with extensive body distribution and multiple physiological functions. Recent studies have witnessed emerging intracellular cascades triggered by APLN and APELA which play crucial roles in female reproductive organs, including hypothalamus-pituitary-gonadal axis, ovary, oviduct, uterus and placenta. However, a comprehensive summary of APLN and APELA roles in physiology and pathology of female reproductive system has not been reported to date. In this review, we aim to concentrate on the general characteristics of APLN and APELA, as well as their specific physiological roles in female reproductive system. Meanwhile, the pathological contexts of apelinergic axis dysregulation in the obstetrics and gynecology are also summarized here, suggesting its potential prospect as a diagnostic biomarker and/or therapeutic intervention in the polycystic ovary syndrome, ovarian cancer, preeclampsia and gestational diabetes mellitus.

## INTRODUCTION

Apelin receptor (APLNR, also known as APJ, APJR, AGTRL1 and HG11) was firstly identified as a class A G protein-coupled receptor in 1993. It consists of 380 amino acids, which has a sequence sharing 31% homology with that of the angiotensin type 1 receptor (*O'Dowd et al., 1993*). Nevertheless, APLNR cannot actually bind to angiotensin II and remains as an ''orphan receptor'' until its endogenous ligand apelin (APLN, also named APEL and XNPEP2) was later extracted from bovine stomach (*Tatemoto et al., 1998*). APLN is generally existed in functional isoforms which are cleaved and modified from the C-terminus of a 77-amino acid pre-pro-peptide encoded by *APLN* gene, with different affinities for APLNR and prevalent distribution (*Chapman, Dupré & Rainey, 2014*). Both

Corresponding authors
Yan Ling, lingyan1205@163.com
Haibin Kuang, kuang-haibin@ncu.edu.cn

APLN and its receptor APLNR levels are universally high at multiple organs like brain, retina, heart, stomach, liver, kidney and blood vessels in many species (*Kawamata et al., 2001*; *Zeng et al., 2007*; *Kasai et al., 2008*; *Qian et al., 2011*; *Krist et al., 2013*; *Lv et al., 2017*). Recent years, apelin receptor early endogenous ligand (APELA, also named ELABELA, Toddler and Ende) was identified as a new endogenous ligand for APLNR in both Chng and Pauli's labs independently (*Chng et al., 2013*; *Pauli et al., 2014*). Similar to APLN, this 54-amino acid polypeptide is also processed into several isoforms. APELA is highly enriched in the early stage of embryo and confirmed to play a vital role in embryogenesis and angiogenesis (*Norris et al., 2017*). APLNR and its two ligands compose the apelinergic axis, which is well delineated in systemic physiological processes like cardiogenesis, angiogenesis, fluid homeostasis, vasodilation and energy metabolism.

More recently, several studies have been investigating the possible intervention of apelinergic axis in female reproductive system based on its precise regulation of steroidogenesis, angiogenesis and vasodilation, before moving onto the dysregulation of this system which hypothetically causes fertility disorders and pregnancy complications like polycystic ovary syndrome (PCOS), ovarian cancer, gestational diabetes mellitus (GDM) and preeclampsia (PE) (summarized in Table 1). This review summarizes and evaluates the current role of apelinergic system in female reproductive system at both physiological and pathological profiles (Fig. 1), as well as providing the direction for future research.

## SURVEY METHODOLOGY

Recently published articles and reports (within 15 years) were conducted from PubMed, Google Scholar and Queen Mary Library databases. Based on the keywords 'APLN', 'APELA' and 'female reproduction', articles extracted were summarized to identify the physiological and pathological roles of apelinergic axis in female reproductive system. This study was approved by Jiangxi Provincial Key Laboratory of Reproductive Physiology and Pathology, Medical Experimental Teaching Center of Nanchang University.

## APLN AND APELA, ENDOGENOUS LIGANDS OF APLNR

### Characteristics of APLN

Human *APLN* gene is located on chromosome Xq25-26.1 which encodes a pre-propeptide of 77 amino acids. After cleavage of the 22-amino acid secretory sequence at N terminus by endopeptidases, the propeptide is subsequently processed into three active fragments at several dibasic residues (Arg-Lys and Arg-Arg), including APLN-36, APLN-17 and APLN-13. APLN-13 undergoes post-transcriptional cyclization at the N-terminal glutamine, generating pyroglutamate-APLN-13 (Pyr1-APLN-13) (*Tatemoto et al., 1998*). The potency and efficacy of APLN differ from different isoforms. For instance, APLN-36, APLN-13 and Pyr1-APLN-13 are preponderantly contributed in human cardiovascular regulation (*Maguire et al., 2009*), whereas APLN-17 plays crucial role in APLNR internalization (*El Messari et al., 2004*). To date, APLN is abundantly distributed in female reproductive system such as ovary, oviduct, uterus and placenta. Emphatically, APLN is identified as one

Wang et al. (2020), *PeerJ*, DOI 10.7717/peerj.10245

**Table 1  Summary of studies about the expressional changes of APLN and APELA in the polycystic ovary syndrome (PCOS), Ovarian cancer (OvCa), preeclampsia (PE) and gestational diabetes mellitus (GDM).**

| Authors | Year | Disease type | Species | Samples | Molecule | No. of cases | No. of controls | Analyzed expression | Significance | Notes |
|---|---|---|---|---|---|---|---|---|---|---|
| Cekmez et al. | 2011 | PCOS | human | plasma | APLN | 48 | 37 | protein | higher in patients ($p < 0.001$) | positive with HOMA-IR |
| Altinkaya et al. | 2014 | PCOS | human | plasma | APLN | 45 | 45 | protein | lower in patients | positive with HOMA-IR |
| Olszanecka-Glinianowicz et al. | 2015 | PCOS | human | plasma | APLN | 87 | 67 | protein | lower in patients | negative with HOMA-IR ($p < 0.001$) |
| Sun et al. | 2015 | PCOS | human | plasma | APLN | 63 | 40 | protein | higher in patients ($p < 0.05$) | positive with HOMA-IR |
| Roche et al. | 2016 | PCOS | human | tissue | APLN and APLNR | 65 | 60 | mRNA and protein | higher in patients ($p < 0.05$) | |
| Bongrani et al. | 2019 | PCOS | human | plasma and tissue | APLN and APLNR | 23 | 27(+28) | mRNA and protein | higher in patients ($p < 0.01$) | |
| Yi et al. | 2017 | OvCa | human | tissue | APELA | NA | NA | mRNA and protein | higher in patients | |
| Neelakantan et al. | 2019 | OvCa | human | tissue | APLNR | NA | NA | mRNA and protein | higher in patients ($p < 0.05$) | |
| Panaitescu et al. | 2018 | EOPE | human | plasma | APELA | 56 | 59 | protein | no difference | |
| Pritchard et al. | 2018 | EOPE | human | plasma | APELA and APLNR | 32 | 32 | mRNA | both no difference | |
| Villie et al. | 2019 | EOPE | human | plasma | APELA | 12 | 14 | protein | no difference | |
| Wang et al. | 2019 | EOPE | human | placenta | APELA | 30 | 30 | mRNA and protein | lower in patients ($p < 0.0001$) | |
| Zhou et al. | 2019 | EOPE | human | placenta | APELA and APLNR | 6 | 11 | mRNA and protein | APELA no difference; APLNR both lower in patients ($p < 0.05$) | |
| Zhou et al. | 2019 | EOPE | human | plasma | APELA | 15 | 15 | protein | no difference | |
| Para et al. | 2020 | EOPE | human | plasma | APELA | 54 | 56 | protein | no difference | |
| Panaitescu et al. | 2018 | LOPE | human | plasma | APELA and APLNR | 57 | 60 | protein | APELA higher in patients ($p = 0.01$); APLNR no difference | |
| Zhou et al. | 2019 | LOPE | human | placenta | APELA and APLNR | 14 | 11 | mRNA and protein | APELA both lower in patients ($p < 0.01$); APLNR protein lower in patients ($p < 0.01$) | |
| Zhou et al. | 2019 | LOPE | human | plasma | APELA | 22 | 15 | protein | lower in patients ($p < 0.01$) | |
| Para et al. | 2020 | LOPE | human | plasma | APELA | 52 | 52 | protein | higher in patients ($p < 0.001$) | |
| Cobellis et al. | 2007 | PE | human | placenta | APLN and APLNR | 15 | 15 | protein | APLN and APLNR expression both higher ($p < 0.05$) | |
| Inuzuka et al. | 2013 | PE | human | placenta | APLN | NA | NA | mRNA and protein | mRNA lower in patients ($p < 0.05$) | |
| Yamaleyeva et al. | 2015 | PE | human | placenta | APLN and APLNR | 20 | 22 | mRNA and protein | APLN lower in patients (only protein $p < 0.05$), APLNR no difference (both mRNA and protein) | |

Wang et al. (2020), *PeerJ*, DOI 10.7717/peerj.10245

**Table 1** (*continued*)

| Authors | Year | Disease type | Species | Samples | Molecule | No. of cases | No. of controls | Analyzed expression | Significance | Notes |
|---------|------|--------------|---------|---------|----------|--------------|-----------------|---------------------|--------------|-------|
| Van Mieghem et al. | 2016 | PE | human | plasma | APLN | 8 | 10 | mRNA | no difference | used antihypertensive treatment |
| Sattar Taha, Zahraei & Al-Hakeim | 2020 | PE | human | plasma | APLN | 60 | 30 | protein | lower in patients ($p < 0.01$) | |
| Aydin | 2010 | GDM | human | breast milk | APLN | 10 | 10 | protein | lower in patients | |
| Telejko et al. | 2010 | GDM | human | plasma | APLN and APLNR | 101 | 101 | mRNA | no difference | |
| Aslan et al. | 2012 | GDM | human | plasma and cord blood | APLN | 30 | 30 | protein | higher in patients ($p = 0.001$) | |
| Boyadzhieva et al. | 2013 | GDM | human | plasma | APLN | 127 | 109 | protein | lower in patients ($p = 0.009$) | |
| Oncul et al. | 2013 | GDM | human | plasma and cord blood | APLN | 24 | 21 | protein | no difference | |
| Akinci et al., | 2014 | GDM | human | plasma | APLN | 141 | 49 | protein | lower in patients ($p < 0.001$) | |
| Caglayan | 2016 | GDM | human | plasma | APLN | 20 | 20 | protein | higher in patients ($p < 0.001$) | |

**Notes.**

For each study, the authors, year, disease type, species, samples, molecule, sample size (No. of cases and controls), analyzed expression and significance were listed. There would be a significance when $p < 0.05$, and $p$ values were listed in the table (if given). The table was ordered by diseases, molecules and year of publication.

PCOS, polycystic ovary syndrome; OvCa, ovarian cancer; PE, preeclampsia; GDM, gestational diabetes mellitus; NA, not available.

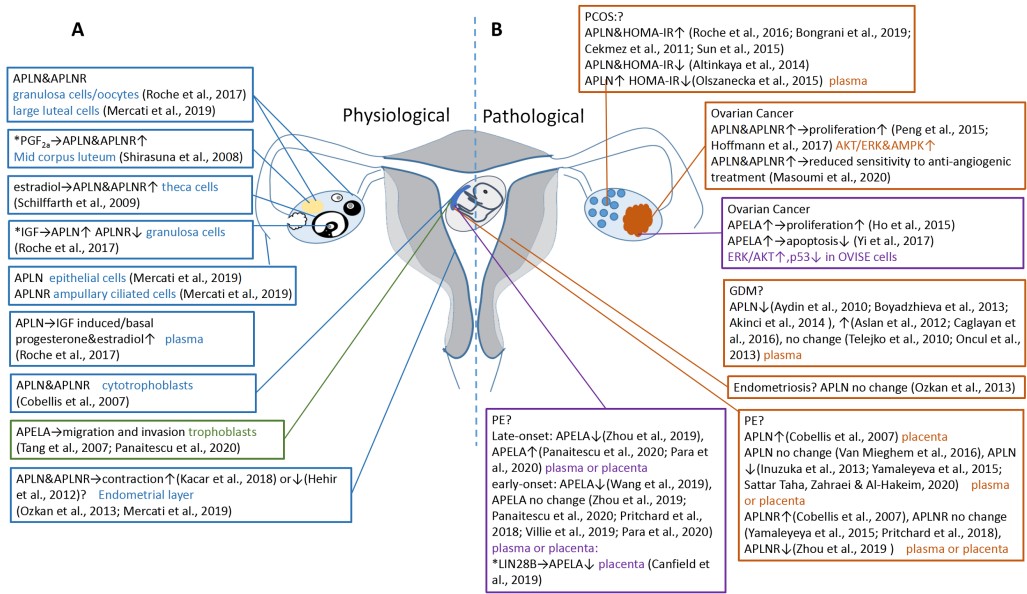

**Figure 1** **Expression and function of APLN and APELA in reproduction system.** (A) In physiological conditions, APLN (in blue textbox) and APELA (in green textbox) play diverse roles at the different parts of the ovary, uterus and placenta. (B) In pathological conditions, aberrant expression of APLN (in orange textbox) and APELA (in purple textbox) lead to female reproductive disorders such as polycystic ovary syndrome (PCOS), ovarian cancer, preeclampsia (PE), gestational diabetes mellitus (GDM) and endometriosis. * indicating potential apelinergic stimulating factors. ? indicating still unclear or controversy about the functions or contributions of apelinergic molecules in these diseases. ↑ and ↓ in the textbox means increase and decrease respectively, both indicate statistically significant changes. → means resulting.

type of adipokines secreted by white adipose tissue, which plays a role with other adipokines in regulating the secretion of gonadotropin releasing hormone (GnRH), gonadotropins and steroids through hypothalamo-pituitary-gonadal (HPG) axis (*Bertrand, Valet & Castan-Laurell, 2015*; *Yang et al., 2019*).

## APLN dependent signaling pathway

APLN/APLNR activates different types of G protein and further stimulates three important signaling pathways, which are phosphorylation of phosphoinositide 3-kinase/protein kinase B (PI3K/Akt), reduction of cyclic adenosine monophosphate (cAMP) and activation of phospholipase C-$\beta$ (PLC-$\beta$), respectively (Fig. 2A) (*Chapman, Dupré & Rainey, 2014*). There are two types of pertussis toxin-sensitive G$\alpha$ protein (G$\alpha$i/o, G$\alpha$q/11) at the downstream of APLNR, mediating different signaling transduction (*Masri et al., 2002*). G$\alpha$i/o activates PI3K/Akt dependent manner which is crucial for cell survival and nitric oxide (NO) induced vasodilation (*Liu et al., 2010*). Akt phosphorylates Bcl-2-associated death promoter (Bad, a BH3-only protein) and shifts it to an inert form, which inhibits the binding of Bad and Bcl-2. Bcl-2 plays an anti-apoptotic role by disturbing the aggregation of Bak and Bax (BH123 proteins) in the mitochondrial outer membrane, and thereby attenuating the release of cytochrome c and activation of caspase-3 (*Liu et al., 2019*).

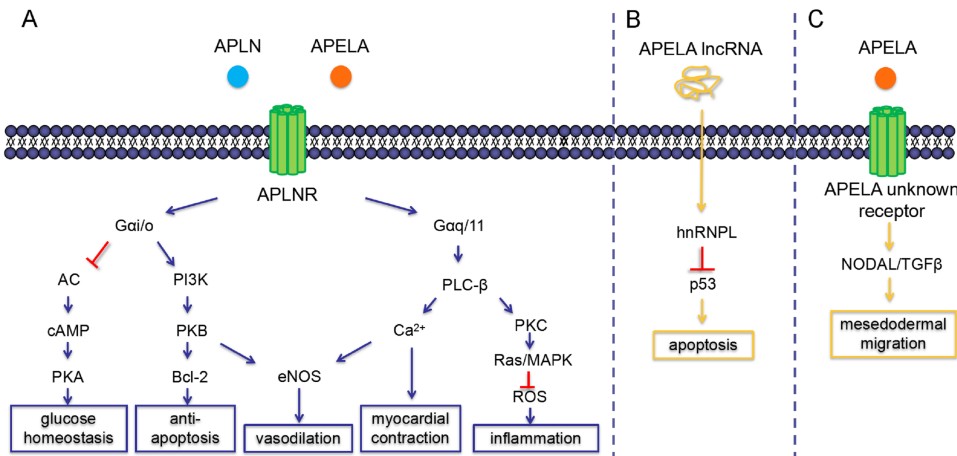

**Figure 2** **Intracellular signaling pathways and physiological functions of APLN and APELA.** (A) Both APLN (in blue) and APELA (in orange) can classically activate Gαi/o and Gαq/11 mediated intracellular transduction via binding to their common receptor APLNR. (B) Non-coding APELA binds to hnRNPL and promotes p53-mediated cell apoptosis. (C) APELA also stimulates PI3K-independent NODAL/TGFβ signal through alternative receptors in hESCs. AC, adenylate cyclase; eNOS, endothelial nitric oxide synthase; IncRNA, long non-coding RNA; hnRNPL, heterogeneous nuclear ribonucleoprotein L.

Moreover, endothelial nitric oxide synthase (eNOS) can also be activated by Akt through phosphorylation, triggering the release of NO for vasodilation (Fig. 2A) (*Yang et al., 2014*). Additionally, Gα-i/o inhibits adenylate cyclase (AC), following with the reduction of 3′, 5′-cAMP and protein kinase A (PKA), which could potentially regulate the glucose homeostasis (Fig. 2A) (*Masri et al., 2002*). Gαq/11 activates PLC-β hydrolyze phosphatidylinositol 4, 5-bisphosphate into second messengers diacylglycerol and inositol trisphosphate, which increases the release of calcium ($Ca^{2+}$) from intracellular store and activates protein kinase C (PKC) (*Carpéné et al., 2007*). Amplified intracellular $Ca^{2+}$ not only mediates positive inotropic effect in cardiac smooth muscle, but also stimulates NO release in periphery via activating eNOS by calmodulin (*Dai, Ramirez-Correa & Gao, 2006*). PKC in turn activates Ras/MAPK system, which plays a crucial role in cell proliferation (*Szokodi et al., 2002*). Furthermore, MAPK halts the expression of pro-oxidant enzymes and subsequently attenuates the release of reactive oxygen species (ROS), which suppresses lipid metabolism and inflammatory reaction (Fig. 2A) (*Than et al., 2014*).

## Characteristics of APELA

*Apela* gene, located on chromosome 4 of *Homo sapiens* (*Fagerberg et al., 2014*) (chromosome 8 in *Mus musculus* (*Yue et al., 2014*) and chromosome 1 in *Danio rerio* (*Ulitsky et al., 2011*), was originally annotated to be transcribed exclusively into a non-coding RNA in zebrafish embryo (*Chng et al., 2013*), while recently it was confirmed to encode a 54-amino acid precursor which further undergoes proteolysis and generates four mature isoforms: APELA-32, APELA-22, APELA-21 and APELA-11. The shortest isoform is conserved across vertebra species (*Huang et al., 2017*). Compared with APLN, APELA as the second discovered endogenous ligand of APLNR is also ubiquitously detected in

placenta, heart, kidney, prostate and mammalian plasma but not as widely as APLN (*Wang et al., 2015*). In addition, studies have reported that APELA is highly expressed in human embryonic stem cells (hESCs) where the APLNR is absent, indicating the existence of an alternative APLNR-independent transduction (*Ho et al., 2015*). A recent report has suggested that an orphan G protein-coupled receptor 25 (GPR25), associated with blood pressure regulation and autoimmune disease, could be activated by both APLN and APELA in non-vertebrates, which is similar as APLN in decreasing the intracellular cAMP level. However, the accurate role of this putative receptor in vertebrate remains to be determined (*Zhang et al., 2018*).

### APELA dependent signaling pathway

Similar to APLN, APELA binds to APLNR, subsequently activating G$\alpha$i/o and G$\alpha$-q/11 mediated signaling cascades, including PI3K/Akt, PKC and PKC-independent Ras/MAPK pathways (Fig. 2A) (*Perjés et al., 2016*; *Zhang et al., 2018*). Nevertheless, it also plays non-redundant role for its specific properties during embryo development. In mouse ESCs, *Apela* acts as a long non-coding RNA that binds to heterogeneous nuclear ribonucleoprotein L (hnRNPL) at the 3′ UTR, which negatively regulates the interaction between p53 and hnRNPL, and promotes p53-mediated DNA damage induced apoptosis (Fig. 2B) (*Li et al., 2015*). In hESCs, APELA acts as a paracrine secreted hormone that binds to an alternative unknown receptor (non-APLNR) and activates PI3K/AKT signaling for cell survival and self-renewal. This process resembles other fibroblast growth factor like exogenous insulin and endogenous insulin-like growth factors (IGFs) in PI3K-mediated cell proliferation. However, APELA-pulsed hESCs is non-redundant as it also implicates in mesendodermal linage commitment through a PI3K-independent manner (*Ho et al., 2015*). During zebrafish gastrulation, a proper level of APELA acting as a mitogen, indirectly mediates the internalization of ventrolateral mesendodermal cells. This process is presumably achieved via activating NODAL/TGF$\beta$ signaling pathway (Fig. 2C), whereas its specific mechanism remains unknown (*Pauli et al., 2014*).

## ROLES OF APLN AND APELA IN HPG AXIS

Endocrine function of female reproduction initiates from hypothalamic GnRH neurons, which mainly receipt projections from arcuate, paraventricular, supraoptic and medial preoptic nuclei of hypothalamus. These neurons secrete GnRH in a pulsatile manner that favours the secretion of lutenizing hormone (LH) and follicle stimulating hormone (FSH) from gonadotroph cells in the anterior pituitary (*Jin & Yang, 2014*). APLN and its receptor are intensively detected in the same nucleic group of the hypothalamus (*Pope et al., 2012*), indicating an essential behavior of them in reproductive regulation. It was reported in both intracerebroventricular and intraperitoneal infusion that APLN-13 suppressed the secretion of FSH and LH in frontal hypophysis in rats, but it cannot cause a disturbance at the GnRH level (*Taheri et al., 2002*; *Tekin et al., 2017*). The structural and functional similarities between APLN and GnRH (*Cho et al., 2007*) were reasonably suspected that APLN could be a competitive inhibitor in the adenohypophysis for GnRH receptors. In addition, the central action of APELA in hypothalamus was also demonstrated that

it exerted as an anorexigenic hormone via binding to APLNR and activating arginine vasopressin and corticotropin releasing hormone neurons in the paraventricular nuclei (*Santoso et al., 2015*). Whether it has an effect on reproductive-dependent hormone or not remains to be illuminated.

## ROLES OF APLN AND APELA IN UTERINE APPENDAGES

### Distribution and role of APLN in the ovarian follicle

Under the normal physiological states, APLN has been identified as a steroidogenic regulator in ovaries of various species including bovine, rat, porcine, sheep and human (*Roche et al., 2016*; *Roche et al., 2017*; *Shuang et al., 2016*; *Rak et al., 2017*; *Mercati et al., 2019*). In cultured bovine follicles, it was firstly reported that *APLN* mRNA was not found in granulosa cells (GCs), whereas *APLNR* mRNA was detected and significantly stimulated by estradiol and progesterone in GCs of estrogen-inactive follicles. In interstitial theca cells, both *APLN* and its receptor mRNA were obviously expressed (*Shimizu et al., 2009*). Two years later, another group cultured bovine ovarian follicles at the similar condition, and found that estradiol over 5 ng/ml (evaluation for follicular maturation) stimulated the expression of APLN and APLNR in theca cells. However, it had no significant effect on the expression of APLN and APLNR in GCs (*Schilffarth et al., 2009*). Recent research confirmed that the expression level of APLN and APLNR were up-regulated in both GCs and oocytes, but remained constant in theca cells (*Schilffarth et al., 2009*; *Shimizu et al., 2009*; *Roche et al., 2017*). In vitro, APLN from GCs of inactive follicles, in response to IGF1 but not to FSH, markedly increased the progesterone production (*Roche et al., 2017*). In porcine follicles, elevated APLN and APLNR were detected following the follicular growth. In turn, APLN significantly increased the secretion of basal steroid hormone (progesterone and estradiol) through the activation of steroidogenic enzyme ($3\beta$HSD and CYP19A1) via AMPK$\alpha$ stimulation, whereas it also decreased the IGF1- and FSH-induced steroid secretion (*Rak et al., 2017*).

### Role of APLN in corpus luteum (CL)

As a potent angiogenic factor, apelinergic axis also plays a role in the transient luteal stage after ovulation. It has been mentioned that this system exclusively exists in the bovine smooth muscle of intraluteal arterioles, with ligands elevated from early to late CL and followed by a significant decrease at regressed CL, while receptors increased from early to mid CL and remained constant till regressed CL (*Shirasuna et al., 2008*). Paradoxically, another study showed that APLNR also decreased significantly after mid CL (*Schilffarth et al., 2009*). Luteolytic factor prostaglandin F2$\alpha$ stimulates APLN and APLNR mRNA expression particularly at the periphery of mid CL (*Shirasuna et al., 2008*). In ewes, both APLN and APLNR proteins were observed in large luteal cells, and the highest level of APLN mRNA was detected in the luteal phase of the ovarian cycle compared to ewes in the anestrous one (*Mercati et al., 2019*). In porcine cultured CL, APLN stimulates $3\beta$HSD activity, which converts inert 5-ene-3 $\beta$HSD to the active 4-ene-3-oxo steroid, therefore it has a pivotal role in progesterone biosynthesis, suggesting an auto/paracrine pattern of the APLN/APLNR system in the ovary (*Rózycka et al., 2018*). In human, this system is found

in the whole ovary through different developmental stages, including luteinized human GCs, theca, oocytes and corona cumulus complex. In cultured luteinized human GCs and follicular fluid, IGF1 exclusively stimulates APLNR expression whereas LH and FSH cannot show the same effect. Conversely, recombinant human APLN-13 and -17 stimulates the secretion of both basal and IGF-induced progesterone and estradiol in a dose-dependent manner, and this process is significantly accelerated in response to IGF1 (*Roche et al., 2016*). This hormone regulation is in agreement to those discovered in bovine that demonstrated APLN could stimulate steroidogenesis and it is speculatively implemented via 3$\beta$ HSD activation and Akt and MAPK3/1 signaling (*Roche et al., 2016*).

## Regulation of APLN in PCOS

PCOS is a common gynecological endocrinopathy characterized by over-expressed LH triggered hyperandrogenism, chronic oligo/anovulation and polycystic ovaries morphology, with clinical manifestations described as "hirsutism, acne, irregular menstruation and subfertility" (*Teede, Deeks & Moran, 2010*; *Teede et al., 2018*). Despite of the positive correlation between PCOS and complications such as visceral obesity, insulin resistance and type 2 diabetes (*Farrell & Antoni, 2010*), the definite aetiology of PCOS at the molecular level still need to be elucidated. It is known that adipokines are bridges to link the energy metabolism and reproductive system, thus they are probably implicated in this process. Hypothetically, APLN controls several aspects of ovarian function in PCOS, underpinned by its role in steroid hormone regulation and insulin resistance. Firstly, the concentration of APLN and its receptors were detected to be significantly increased in PCOS patients with a positive correlation between follicle count and APLN levels (*Bongrani et al., 2019*). This process could be explained by a steroid hormone disturbance effect of APLN in HPG axis. Moreover, as mentioned above, the secretion of APLN in atretic follicles is notably increased in response to IGF1 and insulin, and subsequently stimulates steroidogenesis in GCs (*Boucher et al., 2005*; *Roche et al., 2016*). It indicates a possible implication of insulin in APLN synthesis via activating PI3K/Akt and MAPK3/1 signaling pathways (*Boucher et al., 2005*). Homeostatic Model Assessment for Insulin Resistance (HOMA-IR) and Body Mass Index (BMI) as hall markers of PCOS have been confirmed to be associated with adipocytokines, even if there is still an inconsistency among different researches. In normal cases, studies have revealed either positive or negative correlations of APLN with HOMA-IR and BMI (*Cekmez et al., 2011*; *Olszanecka-Glinianowicz et al., 2015*). In PCOS cases, several groups have shown an enhanced level of serum APLN positively correlated with HOMA-IR and BMI (*Sun et al., 2015*; *Roche et al., 2016*; *Bongrani et al., 2019*), while one research reported a decreased serum APLN level which was positively associated with HOMA-IR and BMI (*Altinkaya et al., 2014*). These discrepant findings among published literature may be attributed to the differences in research design, different stages of PCOS, sample size, genetic characteristics of patients and APLN evaluation methodology.

## Roles of APLN and APELA in ovarian cancer

Previous studies demonstrated that the level of APLN expression was significantly increased in ovarian cancer cells. In MCF-7 cells, the APLN-APLNR system was involved in regulating

the proliferation and metastasis via phosphorylating ERK1/2 pathway (*Peng et al., 2015*). Secretion and expression of APLN as a mitogenic factor was also detected in OVCAR3 cell line which regulates the proliferation progress in a dose-dependent manner (*Hoffmann, Fiedor & Ptak, 2017*). In SKOV3 cell line, over-expressed APLN and its receptor reduced the sensitivity of anti-angiogenic therapeutic regimen (*Masoumi et al., 2020*). Recently, an elevated APELA level was documented in various histotypes of ovarian cancers, especially in ovarian clear cell carcinoma (OCCC) (*Yi et al., 2017*). It is speculated that APELA was involved in multiple pathways in tumorigenesis. For instances, it accelerates cell mitosis and migration through activating ERK and PI3K/AKT cascades (*Ho et al., 2015*). In addition, it was also reported that APELA might negatively regulate p53 in OCCC cell lines, causing non-apoptotic cell growth through an APLNR-independent pathway (*Yi et al., 2017*). However, another study showed that increased APLNR expression was significantly correlated with decreased median overall survival by 14.7 months in patients with high-grade serous ovarian cancer, and APLNR expression was both necessary and sufficient to increase prometastatic phenotypes of ovarian cancer cells including the proliferation, cell adhesion, migration and invasion in vitro (*Neelakantan et al., 2019*).

## Distribution of APLN and APLNR in the oviduct

The expression of apelinergic system in the ovary has been widely discussed. However, currently only one study mentioned its expression in the sheep oviduct. This study showed that APLN was detected in the epithelial cell coat of ampullary ciliated cells, which facilitated the transport of oocytes and spermatozoa through the oviductal tract. APLNR was expressed exclusively in the ampullary secretory cells, suggesting the fertilization and implantation roles of this system during the luteal stage (*Mercati et al., 2019*). In ewe oviduct, the mRNA level of both APLN and APLNR were detected higher in estrus when compared with those in anestrus. As the function of oviduct is to provide place for embryogenesis and transport of early embryo, the disruption of normal oviduct function may cause infertility, which is considering as a serious concern recently, and attracting more expected studies (*Mercati et al., 2019*).

## ROLE OF APLN IN UTERUS

### Distribution and function of APLN and APLNR in uterus

Recently, it has been witnessed that APLN and its receptor also display potential behaviors in uterus among species such as rat, mouse, ewe and human. The expression of APLNR mRNA in uterus was firstly detected through a nonspecific rat tissue RT-PCR screen (*Hosoya et al., 2000*), then its ligand APLN was described to be elevated during the secretory phase in the glandular cells of endometrial layer whereas it remained at a low level in the stromal cells (*Kawamata et al., 2001*; *Ozkan et al., 2013*; *Mercati et al., 2019*). It is evidently deduced that the apelinergic system is stimulated by elevated steroid hormones during the uterine secretory phase also known as the luteal phase of ovarian cycle. APLN subsequently plays a spatio-temporal role in spiral arterioles maturation and interstitial edema in endometrium where angiogenesis is taking place. An in vitro study showed that APLN played a vasodilation role in suppressing both spontaneous and oxytocin-induced

contraction in human myometrial fibers (*Hehir & Morrison, 2012*). However, serum APLN was also reported to exert a positive inotropic effect in rat myometrial layer via PKC-mediated intracellular Ca $^{2+}$ amplication (*Kacar et al., 2018*). These opposite results may be explained by the intracellular balance between vascular dilation and smooth muscle contraction mechanisms of apelinergic system, as well as the impacts of species diversity and reagent concentrations.

### Role of APLN in endometriosis

Endometriosis is defined as an estrogen-dependent invasion of endometrial tissue from uterus to uterine adnexa (*Bulun et al., 2019*). It is a very common cause for chronic pain in the pelvis and could even lead to infertility in moderate and significant patients (Chaljub, Medlock & Services, 2018). Current explanations of endometriosis pathogenesis are endometrial implantation, coelomic metaplasia and induction theories which are all in agreement with the impacts of steroid hormone dysregulation and inflammatory response. Similar expression pattern of APLN was seen in both eutopic and ectopic endometrium during the menstrual cycle indicated that the ectopic endometrial lesion could share some characteristics with eutopic cellular processes in endometrium regeneration (*Mercati et al., 2019*). Additionally, the angiogenesis and vasodilation effects of APLN could potentially be one of the causes in triggering the symptoms of endometriosis, whereas more studies are expected to confirm this point.

## ROLES OF APLN AND APELA DURING PREGNANCY

### Role of APLN and APELA in embryonic development

APLNR was reported to be expressed in the angioblast of frog embryo, which would contribute to the formation of aortic arch vessels and posterior cardinal veins. APLN was detected either within or adjacent to the endothelial cells expressed by APLNR, functioning as an angiogenic agent for nascent blood vessels, especially the intersegmental vessels formation. It also showed the chemotactic ability of APLN to induce the migration of endothelial cells (*Cox et al., 2006*). Moreover, APLN was proved possessing an anti-apoptotic role in osteoblastic cell line of humans and mice (*Tang et al., 2007*; *Xie et al., 2007*). It releases Bcl-2 molecule from Bad via activating PI3K/Akt pathway, which subsequently attenuates the activation of downstream apoptotic factors, such as cytochrome c and caspase-3, resulting in the inhibition of osteoblastic cell apoptosis.

Additionally, APELA has also been revealed to hold a key role in cardiogenesis, angiogenesis and bone formation during the embryonic development. In APELA knockout mice, the hearts are developed poorly or not developed at all, suggesting the essential role in heart morphogenesis (*Chng et al., 2013*). It triggers the endothelial precursor (angioblasts) to migrate towards midline and coalesce underneath the notochord, and form the first axial vessels (*Pauli et al., 2014*; *Helker et al., 2015*). Consistently, APELA-APLNR axis is involved in early placental development and angiogenesis (*Ho et al., 2015*; *Ho et al., 2017*). In mouse placenta, APELA is robustly expressed in syncytiotrophoblasts from early-to-mid gestation, which favors the sprout of new formed blood vessels (*Ho et al., 2017*). It is also associated with skeletal formation through inhibiting the expression of

Sox32, which can bind to Pou5f3 and Nanog molecules as a transcription factor in dorsal endoderm during gastrulation, and inhibit the formation of Pou5f3-Nanog complex. APELA-APLNR pathway can reduce Sox32 expression and allow Pou5f3-Nanog complexes formation, subsequently activating bone morphogenetic protein signaling for sclerotome fate determination (*Perez-Camps et al., 2016*).

## Distribution and function of APLN and APELA in the placenta

Apelinergic system has been hypothesized as a key factor in placental angiogenesis. APLN was strongly expressed in the cytoplasm of human cytotrophoblasts during the first two trimester of pregnancy, and then decreased at the third trimester. Subtle signals were also detected in the syncytiotrophoblasts during the first trimester, but it disappeared completely in the third trimester (*Cobellis et al., 2007*). The expression of APLNR in the placenta was later than that of APLN. In the first trimester, it was relatively low and exclusively in the cytotrophoblasts. However, in the third trimester, APLNR was expressed intensely not only in cytotrophoblasts but also in syncytiotrophoblasts, smooth muscle cells and endothelial cells inside of the placental villi (*Cobellis et al., 2007*). This change suggests a potential chemoarractant and vasculogenic role of APLN in the invasion process of interstitial and endovascular extravillous trophoblasts. In mouse, APELA was detected initially in the trophoblasts and then increased robustly after the allantoic fusion. At the mid-gestation, it was expressed restrictedly in syncytiotrophoblasts, where APLNR was wildly existed in adjacent endothelial cells, indicating a paracrine function of this system to favor the placental angiogenic sprouting (*Ho et al., 2017*). However, in human placenta, APELA was expressed in both cytotrophoblasts and syncytiotrophoblasts synchronously during the whole pregnancy (*Ho et al., 2017*) and its speculated role remains to be illuminated.

## Regulation of APELA and APLN in PE

The basic pathological changes of hypertensive disorders in pregnancy are currently recognized as insufficient spiral arteries recasting and inflammation mediated endothelial damage triggered by the intricate network of signaling cascades. APELA as mentioned above plays a crucial role in placental angiogenesis via activating PI3K/AKT/mTOR pathway (*Ho et al., 2017*), and whether its reduction could lead to PE is now being widely studied. It was firstly discovered that APELA knockout pregnant mice exhibited a hypertensive symptom accompanied with proteinuria and glomerular endotheliosis, which were manifested as preeclampsia-like symptoms (*Ho et al., 2017*). Scientists therefore started investigating the change of APELA in PE patients and wanted to know whether the APELA could act as a biomarker (*Zhou et al., 2019*). In the late-onset PE (LOPE), two studies measured a significant increased concentration of APELA in the placenta and serum (*Panaitescu et al., 2020*; *Para et al., 2020*), while one study measured significant decrease (*Zhou et al., 2019*). And for early-onset PE (EOPE), only one study observed decrease in both APELA mRNA and protein (*Wang et al., 2019*), while other studies report no significant change of APELA level on either protein or mRNA (*Pritchard et al., 2018*; *Villie et al., 2019*; *Zhou et al., 2019*; *Panaitescu et al., 2020*; *Para et al., 2020*). Furthermore, it was found that hypoxia significantly decreased the expression of LIN28B, LIN28A and APELA, and the

downregulation of LIN28B and APELA may play a role in PE by reducing trophoblast invasion and syncytialization (*Canfield et al., 2019*).

There are also contradictions about the expression level of APLN in PE patients. Initially, a clinical study found an increased APLN protein level in the placental samples of PE patients, indicating a speculated correlation between APLN and PE (*Cobellis et al., 2007*). This study was further proved by an experiment which showed intravenous injection of APLN in male mice could lead to the downregulation of blood pressure, suggesting that APLN might act as a vasodilator in PE (*Lee et al., 2000*). However, case studies also found either decreased (*Inuzuka et al., 2013*; *Yamaleyeva et al., 2015*; *Sattar Taha, Zahraei & Al-Hakeim, 2020*) or no significantly changed (*Van Mieghem et al., 2016*) APLN level in PE patients compared with normotensive pregnancies.

Not only its ligands, the expression of APLNR is also rather conflicting. It has long been thought that APLNR level increases when the patient suffer from PE (*Cobellis et al., 2007*), but two other studies suggest APLNR level remains unchanged when PE occurs (*Yamaleyeva et al., 2015*; *Pritchard et al., 2018*). However, one recent study found a significant decrease both in APLNR mRNA expression and in situ expression between PE patients and normal control, and this significance can be found when controls compared to both EOPE and LOPE groups (*Zhou et al., 2019*).

Altogether, the different expression of apelinergic system could be explained by confounding factors like BMI and mean maternal age mismatches between the cases and controls. Moreover, the balance between vasorelaxant and myocardial contractile effects of apelinergic system, as well as the crosslink of apelinergic axis with intricate inflammatory and endothelial factors in PE should also be taken into consideration. Further investigations should focus on the specific molecular mechanisms of APLN and APELA in the hypertensive disorders of pregnancy.

## Regulation of APLN and APELA in GDM

APLN as one of the adipose tissue-derived hormones has been identified to play a role in blood glucose metabolism (*Antushevich & Wójcik, 2018*). It has been described that insulin may up-regulate the expression of APLN through PKC and PI3K signaling pathways in both murine and human adipocytes (*Boucher et al., 2005*). Raised apelin levels were found in both insulin-resistant mice and type 2 diabetes mellitus patients (*Xu, Tsao & Yue, 2011*), which supported the speculation that insulin can stimulate APLN secretion. Nevertheless, the correlation of APLN levels with GDM has not yet reached an agreement in clinical researches. Three studies reported a decrease of serum APLN level in GDM patients (*Aydin, 2010*; *Boyadzhieva et al., 2013*; *Akinci et al., 2014*) while two other groups revealed an increase (*Aslan et al., 2012*; *Kiyak Caglayan et al., 2016*). In contrast, there were also no significant association reports between normal control and GDM patients (*Telejko et al., 2010*; *Oncul et al., 2013*). Thus, the correlation of APLN with the pathophysiology of GDM remains to be elucidated. In addition, factors like BMI, HOMA-IR and birth weight have been shown not correlated with serum APLN level according to these studies (*Aslan et al., 2012*; *Oncul et al., 2013*), but these confounding factors varied a lot among different groups during pregnancy, which was probably one of the cases in the controversy. APELA

had a positive correlation with fasting plasma glucose levels in healthy pregnant women during the second trimester, while decreased APELA circulating level was observed in GDM patients at the same time. In the third trimester, circulating APELA level decreased significantly in both GDM and healthy groups. This study suggested that APELA could be a physiological demand in glucose metabolism, and further contributions should focus on dynamic levels monitoring and mechanism analysis (*Guo et al., 2020*).

## CONCLUSION

This review presents a landscape of the novel APLN/APELA-APLNR system in the female reproductive field (Table 1 and Fig. 1). Intricate signaling pathways and crosslinks of APLN and APELA imply their multifunctional roles in different organs like ovary, uterus and placenta, during specific developmental stages. APLN as an adipokine appears to have specific effects in steriodogenesis and metabolic regulation in GCs and CL of the ovary. Insulin and IGF1-induced APLN secretion possibly plays a role in glucose regulation in GDM patients. In addition, APLN may sustain a balance between the vasodilative and myocontractile effects in the uterus which could be correlated with hypertensive disorders during the pregnancy. Similarly, APELA as a novel ligand of APLNR also has a potential role in PE, based on the angiogenic effect of spiral arterioles. APELA is essential for fetal and placental development through stimulating the invasion of extravillous trophoblasts. This process is potentially achieved through a chemo-attractant mechanism in placental angiogenic sprouting. Moreover, there is a hyperplasia effect of APELA which could be one of the causes in ovarian tumorigenesis.

All the data suggest that there should be additional studies to further investigate the precise roles of this axis in female reproductive system especially at the pathological profile. In the future, it will be important to clarify the crosslink and interaction between APLN and other adipokines in sex hormone regulation and energy metabolism. Specific expression and biological effects of APELA in ovary and uterus are also needed in prospect. It may also be crucial to identify the balance of smooth muscle contraction and vasodilation in apelinergic system at a molecular hierarchy. Collectively, the apelinergic axis is still a novel project for further investigation in both physiological and pathological aspects, and probably brings better therapeutic or prophylactic intervention towards female reproductive disorders.

**List of Abbreviations**

| | |
|---|---|
| **AC** | adenylate cyclase |
| **APELA** | apelin receptor early endogenous ligand |
| **APLN** | apelin |
| **APLNR** | apelin receptor |
| **Bad** | Bcl-2-associated death promoter |
| **BMI** | body mass index |
| **Ca$^{2+}$** | calcium |
| **cAMP** | cyclic adenosine monophosphate |
| **CL** | corpus luteum |
| **eNOS** | endothelial nitric oxide synthase |

| | |
|---|---|
| **EOPE** | early-onset preeclampsia |
| **FSH** | follicle stimulating hormone |
| **GDM** | gestational diabetes mellitus |
| **GnRH** | gonadotropin releasing hormone |
| **GPR25** | G protein-coupled receptor 25 |
| **GC** | granulosa cell |
| **hESC** | human embryonic stem cell |
| **hnRNPL** | heterogeneous nuclear ribonucleoprotein L |
| **HOMA-IR** | homeostatic model assessment for insulin resistance |
| **HPG axis** | hypothalamo-pituitary-gonadal axis |
| **IGF** | insulin-like growth factor |
| **LH** | lutenizing hormone |
| **LOPE** | late-onset preeclampsia |
| **NO** | nitric oxide |
| **OCCC** | ovarian clear cell carcinoma |
| **PCOS** | polycystic ovary syndrome |
| **PE** | preeclampsia |
| **PI3K/Akt** | phosphoinositide 3-kinase/protein kinase B |
| **PKA** | protein kinase A |
| **PKC** | protein kinase C |
| **PLC-$\beta$** | phospholipase C-$\beta$ |
| **ROS** | reactive oxygen species. |

### Funding

This study was financially supported by the National Natural Science Foundation of China (32060203 and 81860283), Funds of Health and Family Planning Commission of Jiangxi Province (20195050) and The 555 project of Jiangxi Province Gan Po Excellence (18000066). The funders had no role in study design, data collection and analysis, decision to publish, or preparation of the manuscript.

### Grant Disclosures

The following grant information was disclosed by the authors:
National Natural Science Foundation of China: 32060203, 81860283.
Funds of Health and Family Planning Commission of Jiangxi Province: 20195050.
The 555 project of Jiangxi Province Gan Po Excellence: 18000066.

### Competing Interests

The authors declare there are no competing interests.

## Author Contributions

- Xueying Wang and Xiaofei Liu conceived and designed the experiments, performed the experiments, analyzed the data, prepared figures and/or tables, authored or reviewed drafts of the paper, and approved the final draft.
- Zifan Song, Xin Shen and Siying Lu performed the experiments, authored or reviewed drafts of the paper, and approved the final draft.
- Yan Ling and Haibin Kuang conceived and designed the experiments, performed the experiments, analyzed the data, authored or reviewed drafts of the paper, and approved the final draft.

## Data Availability

This is a literature review without any raw data.

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
