# Peer review of "Emerging roles of APLN and APELA in the physiology and pathology of the female reproductive system"

_PeerJ, doi:10.7717/peerj.10245_

## Round 0.1 · original submission · Major Revisions

This manuscript is of interest. However, it needs to be deeply improved. The main concern is about references; since this review is intended to describe the apelin system in the female reproductive apparatus, all information existing in the scientific literature must be considered.

Reviewer 1 ·

Basic reporting

No comment

Experimental design

Here there are some suggestions related to the sources cited, the topics and paragraphs considered.

Line 96: delete “(1996)”. The year is already write in the citation.
Line 134: please, check the reference “Chun et al., 2010” and verify that it is correct in this position. Information described in the previous sentences (From line 129 to line 134: Human APLN gene is located on ….. …. generating pyroglutamate-APLN-13 (Pyr1-APLN-13) cannot be found in Chun et al., 2010.
Line 138. Since the review takes into account the entire female reproductive system and not just the ovaries, the oviducts should also be added in the following sentence: "To date, APLN is widely distributed in the female reproductive system such as the ovary, uterus and placenta". Please, consider: Mercati F, Scocco P, Maranesi M, et al. Apelin system detection in the reproductive apparatus of ewes grazing on semi-natural pasture. Theriogenology. 2019;139:156-166. doi:10.1016/j.theriogenology.2019.08.012"
Line 142. It would be better to insert at least another citation describing not only apelin system in energy metabolism but also its role in GnRH, gonadotropins and steroids through hypothalamo-pituitary-gonadal axis; for example, Yang N, Li T, Cheng J, Tuo Q, Shen J. Role of apelin/APJ system in hypothalamic-pituitary axis. Clin Chim Acta. 2019;499:149-153. doi:10.1016/j.cca.2019.09.011.
Line 228. Please, check the reference “Kwak et al., 2019”. Information regarding “APLN as a steroidogenic regulator in ovaries” are not included in this article.
In chapter 5 "Roles of APLN and APELA in the ovary", sheep must also be mentioned among the species in which apelin and apelin receptor were detected at the follicle and corpus luteum level, as described in “Mercati et al. Theriogenology. 2019;139:156-166. doi:10.1016/j.theriogenology.2019.08.012"
Since this review is intended to describe the apelin system in the female reproductive apparatus, all information existing in the scientific literature must be considered. Apelin and apelin receptor have been described in the ewe oviduct with a higher level of apelin during the luteal phase than in the anestrous phase of the ovarian cycle. A role of the apelin system in the regulation of gamete transport and nutrition and support along the oviduct has been hypothesized. I suggest that the data on the oviduct are taken into account and are described in a dedicated paragraph. Please, see this publication: “Mercati F, et al. Theriogenology. 2019;139:156-166. doi:10.1016/j.theriogenology.2019.08.012

Validity of the findings

No comment.

Additional comments

Dear Authors,
This review deals with a very interesting and current topic. As the reviewer’s knowledge, to date there are not papers that describe apelin system on all organs and structures of female genital system as it is showed in this review.
The manuscript is clearly written and English style is fluent. A number of articles have been taken into account by Authors. Each section is accurately described and well detailed. However, a paragraph describing apelin on oviduct should be added.
I have made some corrections in order to improve the clarity of the text and enrich information where necessary.
All consideration are listed below:

Line 91: “HG11” should be mentioned among receptor known names.
Line 96: delete “(1996)”. The year is already write in the citation.
Line 134: please, check the reference “Chun et al., 2010” and verify that it is correct in this position. Information described in the previous sentences (From line 129 to line 134: Human APLN gene is located on ….. …. generating pyroglutamate-APLN-13 (Pyr1-APLN-13) cannot be found in Chun et al., 2010.
Line 138. Since the review takes into account the entire female reproductive system and not just the ovaries, oviducts should also be added in the following sentence: "To date, APLN is widely distributed in the female reproductive system such as the ovary, uterus and placenta". Please, consider Mercati F, Scocco P, Maranesi M, et al. Apelin system detection in the reproductive apparatus of ewes grazing on semi-natural pasture. Theriogenology. 2019;139:156-166. doi:10.1016/j.theriogenology.2019.08.012"
Line 142. It would be better to insert at least another citation describing not only apelin system in energy metabolism but also its role in GnRH, gonadotropins and steroids through hypothalamo-pituitary-gonadal axis; for example, Yang N, Li T, Cheng J, Tuo Q, Shen J. Role of apelin/APJ system in hypothalamic-pituitary axis. Clin Chim Acta. 2019;499:149-153. doi:10.1016/j.cca.2019.09.011.
Line 172. Delete a parenthesis: “(Ulitsky et al., 2011))”
Figure 2. In reviewer’ opinion Apela is (in yellow) and not (in orange).
Line 195. Figure 2 C should be mentioned after figure 2 B that is at line 203.
Line 228. Please, check the reference “Kwak et al., 2019”. Information regarding “APLN as a steroidogenic regulator in ovaries” are not included in this article.
In chapter 5 "Roles of APLN and APELA in the ovary", sheep must also be mentioned among the species in which apelin and apelin receptor were detected at the follicle and corpus luteum level, as described in “Mercati F, Scocco P, Maranesi M, et al. Apelin system detection in the reproductive apparatus of ewes grazing on semi-natural pasture. Theriogenology. 2019;139:156-166. doi:10.1016/j.theriogenology.2019.08.012"
Since this review is intended to describe the apelin system in the female reproductive apparatus, all information existing in the scientific literature must be considered. Apelin and apelin receptor have been described in the ewe oviduct with a higher level of apelin during the luteal phase than in the anestrous phase of the ovarian cycle. A role of the apelin system in the regulation of gamete transport and nutrition / support along the oviduct has been hypothesized. I suggest that you consider the data on the oviduct described in this publication: “Mercati F, Scocco P, Maranesi M, et al. Apelin system detection in the reproductive apparatus of ewes grazing on semi-natural pasture. Theriogenology. 2019;139:156-166. doi:10.1016/j.theriogenology.2019.08.012
Line 322: replace “adema” with “edema”.
Line 432: replace “marine” with “murine”.
Table 1. In the title of the table, it would be better to write the full name of disease type than their abbreviation.
Figure 1. In the caption, please, explain the significance of the arrows contained in textboxes.
Please, check English language.

Reviewer 2 ·

Basic reporting

no comment

Experimental design

no comment

Validity of the findings

no comment

Additional comments

Apelin and Apela are two endogenous ligands of APLNR, and composing the apelinergic axis with APLNR. The apelinergic pathway has been generating increasing interest in the past few years for its potential as a therapeutic target in heart failure, pulmonary arterial hypertension, atherosclerosis, but also type 2 diabetes, and preeclampsia. This review summarizes and evaluates the current role of apelinergic system in female reproductive system at both physiological and pathological conditions. suggesting its potential prospect as a diagnostic biomarker and/or therapeutic intervention in the polycystic ovary syndrome, ovarian cancer, preeclampsia and gestational diabetes mellitus. This work is of significance to Gynaecology and Obstetrics.

1、Recent published paper in J Matern Fetal Neonatal Med (PMID: 32008387) should be cited in this review.
2、The description of APLNR in PE should be added.

3、The expression level of APLA in patients with late-onset PE is controversial (Ho et al., 2017 ,Wang et al., 2019; Zhou et al., 2019, Canfield et al., 2019) , while that of ELA in patients with early-onset PE is consistent (Pritchard et al., 2018; Villie et al., 2019; Zhou et al., 2019; Panaitescu et al., 2020), Therefore, table1 should distinguish early-onset PE from late-onset PE.

4、Please, add corresponding reference(s) the following sentences in 3.2: ……reduction of cyclic adenosine monophosphate (cAMP) and activation of phospholipase C-β (PLC-β), respectively (Fig 2 A).
Gαi/o activates PI3K/Akt dependent manner which is crucial for cell survival and nitric oxide (NO) induced vasodilation.

---

## Round 0.2 · Minor Revisions

Thank you for improving your paper. After a careful review it is needed to do some modifications in the references. Please, check all of them and making consistent (more comments are provided by reviewer number 1).

Reviewer 1 ·

Basic reporting

English revision is suggested.

Experimental design

No comment

Validity of the findings

No comments

Additional comments

The manuscript entitled “Emerging roles of APLN and APELA in physiology and pathology of female reproductive system” was edited as suggested and a point by point response to my comments were provide. However, I have a few other considerations to address.


Figure 2: The figure and the text were changed as suggested but the caption were not changed. Please, reverse B and C description.
Line 101: Change “and” with “&” in this reference and other references in the text.
Line 234-235: Since Kwak reference was delete, also “mouse” should be deleted by the sentence. Otherwise add another reference regarding mouse species.
Line 488: Please, change “another two groups” with “two other groups”.
Line 501: “Guo et al., 2020” is missing in the references chapter.
Line 637: Please, do not delete “92(6):431-440.”
I suggest to revise citations in the text and in the chapter of References where several references were cancelled even if they are cited in the text. For example: Fagerberg et al., 2014; Lee et al., 2000; Than et al., 2014; Wang et al., 2019;

Reviewer 2 ·

Basic reporting

no comment

Experimental design

no comment

Validity of the findings

no comment

Additional comments

The author answered all my questions and I agreed to accept. however, there is one point that should be corrected by the authors.
In table 1, Zhou et al. also measured ELA concentrations in EOPE (n=15) and LOPE (n=22) using plasma samples, and the sample size used was larger than that of placental tissue. Therefore, the results are more convincing and should be reflected in Table 1.

---

## Round 0.3 · accepted · Accept

Dear Authors,
I am pleased to confirm that your paper has been accepted for publication in PeerJ.

Please ensure that the error noted by Reviewer 1 is addressed during production.

Thank you for submitting your work to this journal.

Reviewer 1 ·

Basic reporting

No comment

Experimental design

No comment

Validity of the findings

No comment

Additional comments

The manuscript has been edited and improved from the initial version.
I have no other comments to add except that one of the suggested changes was not made as stated in the rebuttal letter. Probably due to an oversight of the authors. Please, note that the noun “mouse” was not deleted at line 234.

Reviewer 2 ·

Basic reporting

no comment

Experimental design

no comment

Validity of the findings

no comment

Additional comments

no comment